# A trend analysis of Black American women with cardiovascular disease and chronic medical conditions, sociodemographic factors from NHANES From 2011 to 2020

**Rezaul Karim Ripon**[1,2,3]*, **Fahmida Hoque Rimti**[4], **Mickelder Kercy**[5], **Shahriar Hossain**[2,3], **Umma Motahara**[2], **Md. Sharif Hossain**[3,6], **Md. Tajuddin Sikder**[2]

**1** Harvard T.H. Chan School of Public Health, Boston, Massachusetts, United States of America, **2** Department of Public Health and Informatics, Jahangirnagar University, Savar, Dhaka, Bangladesh, **3** McHigher Center for Health Research, Boston, Massachusetts, United States of America, **4** Chittagong Medical College, Chittagong, Bangladesh, **5** The City University of New York, New York, New York, United States of America, **6** Division of Epidemiology, Institute for Health and Equity, Medical College of Wisconsin, Milwaukee, Wisconsin, United States of America

* riponrezaul5@gmail.com

## Abstract

### Background

Significant racial and gender differences exist in the prevalence of CVD in the United States. The goal of this study is to evaluate the prevalence of CVD among Black American women, the relationship between CVD and some medical conditions, and significant sociodemographic factors.

### Methods

The researchers in this study used data from four cycles of the NHANES, carried out by the NCHS. 2011 to 2012, 2013 to 2014, 2015 to 2016, and 2017 to 2020 were the cycles that were chosen. The researchers used the survey package in the R programming language to examine the data.

### Results

People with CVD problems 20 years of age and older were included in the analyses. Black American women experienced a considerable prevalence of CVD from 2011 to 2020. These women were more likely to report having completed no more schooling than the ninth grade, being widowed/divorced/separated, and having undergone a hysterectomy, as well as having a history of diabetes, asthma, obesity, arthritis, and depression. Black American women with CVD had a 3.8-fold increased risk of diabetes and a 5.6-fold increased risk of arthritis.

### Conclusion

This study shows that Black American women with CVD are more likely to have chronic illnesses such as hysterectomy, diabetes, asthma, obesity, arthritis, and depression. Black

**Data Availability Statement:** The dataset is publicly available in http://www.cdc.gov/nchs/nhanes.htm.

**Funding:** The authors received no specific funding for this work.

**Competing interests:** The authors have declared that no competing interests exist.

American women's cardiovascular risk profiles can be updated using the data from this study.

## Introduction

CVD and other NCD are the leading causes of high and premature mortality in high- and low-income countries [1]. Comparing developed countries, the burden of NCD remains the same or has been increasing in the United States, particularly among women [2]. Compared to other racial and ethnic groups, Black American women have been found to have higher rates of CVD, other non-communicable chronic medical problems, and worse outcomes [3]. To address these discrepancies, it can be helpful to understand the trends in CVD and other chronic illnesses among Black American women [4]. In the United States, CVD is the primary cause of mortality for women [4]. Non-communicable chronic illnesses, including diabetes, asthma, and arthritis all significantly increase the burden of illness [3]. Public health professionals can better allocate resources for their management and prevention by looking at trends in these disorders. Chronic diseases like CVD, diabetes, and arthritis can have serious long-term health effects that can result in disability, a decline in quality of life, and a rise in medical expenses [5]. The long-term health outcomes for Black American women can be improved through public health policies and interventions [5], which are informed by trends in these conditions.

Previous studies have indicated that Black American women are more likely than other demographic groups to experience cardiovascular disease and other chronic medical disorders [3, 5]. Socioeconomic position, access to healthcare, genetics, and lifestyle elements including nutrition and physical activity levels all may play a role in this discrepancy [6]. Other determinants of these health conditions include high-quality food insecurity [7, 8], neighborhood crime [9], low educational attainment and low-income status according to ethnicity [10], systemic socio-political, economic, and environmental disparities, in addition to the role of gender [11]. The high incidence, prevalence, morbidity, and mortality rates in CVD and other NCD among Black Americans that start at birth and last throughout their lifespan until death are caused by these social determinants of health linked to systemic chronic inflammation [12]. Black American women are disproportionately more affected by these health disparities and inequities, as the literature indicates [13, 14]. Cardiovascular and other non-communicable diseases account for the majority of healthcare spending in the United States ($255.1 billion, 95% UI: 233-4-282-6), with five modifiable health risk factors (high body mass index, high systolic blood pressure, high fasting plasma glucose, dietary risks, and tobacco smoke) accounting for 70% (95% UI: 65–74) of this estimate. Additionally, estimates place the costs of providing informal care for patients with cardiovascular disease in the U.S. between $61 billion in 2015 and $128 billion in 2035 [15].

According to DeSalvo et al. (2016), research on socioeconomic determinants of health and how they affect health outcomes is desperately needed, especially for populations who are more likely to experience health disparities [16]. Moreover, looking into trends in sociodemographic factors over the previous ten years may help in identifying areas for improvement and informing targeted interventions to reduce health disparities given the ongoing COVID-19 pandemic and its impact on health outcomes, including cardiovascular disease and chronic medical conditions [17]. Overall, the importance of this study issue may be seen in its potential to shed light on the patterns of CVD and other chronic medical conditions among Black American women, which can help guide plans for enhancing health outcomes and eradicating

health disparities in this community. This study aims to investigate the current trend prevalence of CVD and quantify its association with other medical conditions and risk factors among Black American women to address better health disparities and inequities in clinical and population health prevention, management and outcomes.

## Methodology

### Study population and design

NHANES, an ongoing cross-sectional study that aims to estimate the prevalence and distribution of various diseases and risk factors in the United States, is managed by the Centers for Disease Control and Prevention's National Center for Health Statistics. NHANES oversamples a variety of distinct demographic groups, including those who are Hispanic, non-Hispanic Black, non-Hispanic Asian, older adults, and low-income persons, in order to offer appropriate samples for subgroup analysis and more precise variance estimates. Only data from Black American individuals in the NHANES 2011–2020 survey were used in our analysis [18]. Age, family income to poverty, country of birth, education level, marital status, vigorous work activity, and moderate work activity were among the demographic details that participants self-reported. A doctor confirmed that she had undergone a hysterectomy, diabetes, asthma, obesity, arthritis, cancer, and depression. All participants provided written informed consent for the study, which received approval from the NCHS institutional review board [18]. All methods were conducted in accordance with the ethical standards of the declaration of Helsinki.

### Data collection and measures

The study team interviewed participants at their homes to gather data on their age, family income to the poverty level, country of birth, degree of education, marital status, and level of vigorous and moderate work activity. Additionally, during a consultation held at the examination center, doctors confirmed the existence of chronic medical conditions such as hysterectomy, diabetes, asthma, overweight, arthritis, cancer, and depression.

### Measures (dependent variable)

Four NHANES cycles—2011–2012, 2013–2014, 2015–2016, and 2017–2020—were used to collect the data we used [18]. Our study's population of Black American women with CVD was the dependent variable. Any affirmative response connected to any of the following variables was used to define CVD: MCQ160a (Ever told you had congestive heart failure), MCQ160b (Ever told you had congestive heart failure), MCQ160c (Ever told you had coronary heart disease), MCQ160d (Ever told you had angina/angina pectoris), MCQ160e (Ever told you had a heart attack), and MCQ160f (Ever told you had a stroke).

### Other measures (independent variables)

The study used a range of independent variables to examine their relationships to chronic illnesses like CVD and others in Black American women. Age, family income to poverty ratio, place of birth, level of education, married status, level of vigorous and moderate work activity, hysterectomy, diabetes, asthma, overweight, arthritis, cancer, and depression were among these factors. The RIDRETH1 and RIAGENDR variables were utilized to represent race and gender, respectively. RIDAGEYR was used to calculate age, INDFMPIR to calculate family income, DMDEDUC2 to calculate education, and DMDMARTL to calculate marital status. DMDBORN4 was used to establish the birthplace, PAQ605 was used to measure strenuous work activity, and PAQ620 was used to determine moderate work activity. Diabetes was

detected using DIQ010, hysterectomy by RHD280, asthma by MCQ010, obesity by MCQ080, stroke by MCQ160f, cancer by MCQ220, and depression by the Patient Health Questionnaire, a nine-item depression screening tool that incorporates DSM-IV diagnostic criteria. Scores over 5 on the measure, which employs a point system based on responses to each item, signify a depressed condition [19].

## Statistical analysis

Oversampling, stratification, and clustering are some of the complicated survey design techniques used by the NHANES. Each participant is given a weight to account for varying sampling likelihood and nonresponse. In this study, for each two-year survey cycle from 2011 to 2020, we computed the prevalence (%) of CVD among Black Americans. And age, ratio of family income to poverty, country of birth, education level, marital status, vigorous work activity, moderate work activity, hysterectomy, diabetes, asthma, overweight, arthritis, cancer, and depression stratified by CVD prevalence, using survey methodologies for all analyses. We addressed the complex survey design, including oversampling, survey nonresponse, and post-stratification, using the 2-year weights computed using the NCHS technique. SURVEYFREQ and SURVEYREG were two methods we used to analyze the data [18, 20–23]. We conducted an analysis using SURVEYFREQ to determine the prevalence of various responses and identify patterns or trends within the data. The SURVEYREG method was employed for the regression analysis. To examine linear trends over survey cycles and calculate p-values for trends, we used weighted logistic regression. In the regression models, the survey cycle was treated as a continuous variable. Missing data were not included in the study. The programming language R was employed to carry out the investigation. P<0.05 (two-sided) was used to determine statistical significance [18, 24–28].

## Results

The Table 1 shows the prevalence percentages of CVD and numerous other variables among Black American women over the course of various years. The total prevalence of CVD rose from 7.92% in 2013–2014 to 10.9% in 2015–2016 before dipping slightly to 10.2% in 2017–2020 (P for trend = 0.03). The participants' mean age grew from 43.22 to 46.27 years between 2011 and 2020 (P for trend = 0.053). Over the course of the study, there was no discernible trend in the proportion of family income to poverty (P for trend = 0.435). Although the prevalence of CVD among Americans born between 2011 and 2012 and 2017 and 2020 increased, this trend was not statistically significant (P for trend = 0.543). According to education level, CVD prevalence was highest among those with less than a ninth-grade education (41%) and lowest among those with a college degree or higher (7%) in the same year. Statistics showed that there were disparities in ninth-grade education of education (P = 0.0004). The prevalence of CVD varied significantly depending on marital status as well. In the years between 2017 and 2020, individuals who were widowed/divorced/separated had the highest prevalence of CVD (32%) while those who were married or cohabiting had the lowest prevalence (9%). There were statistically significant variations in the groups according to marital status (P = 0.014). Throughout the study period, there were statistically significant differences between groups in the prevalence of hysterectomy, diabetes, asthma, arthritis, and depression. For instance, the prevalence of hysterectomy rose from 27% to 25% between 2011 and 2020 (P = 0.001), while the prevalence of asthma fell from 15% to 13% between 2011 and 2020 (P = 0.004). With the highest prevalence in 2015–2016 (14%) and the lowest prevalence in 2013–2014 (6%), the prevalence of overweight also varied throughout the course of the study period and revealed a

**Table 1. Prevalence % (95% CI) of CVD among Black American women.**

| | | 2011–2012 | 2013–2014 | 2015–2016 | 2017–2020 | P for trend |
|---|---|---|---|---|---|---|
| Overall CVD | | 10.7(8–14) | 7.92(5–11) | 10.9(6–15) | 10.2(5–15) | .03 |
| Age | Mean (95%CI) | 43.22(40.79–45.65) | 44.41(41.94–46.86) | 45.11(42.72–47.49) | 46.27(44.53–48) | .053 |
| Ratio of family income to poverty | Mean (95%CI) | 2.11(1.62–2.60) | 2.02(1.81–2.23) | 2.2(1.87–2.53) | 2.2(1.93–2.47) | .435 |
| Born in US | Yes | 11(8–14) | 8(5–12) | 12(7–17) | 14(6–20) | .543 |
| Education Level | Less than 9th Grade | 15(3–34) | 8(10–26) | 10(11–30) | 41(5–76) | 0.0004 |
| | 9-11th Grade (Includes 12th grade with no diploma) | 22(13–30) | 14(6–22) | 17(7–28) | 18(8–28) | N/A |
| | High School Grad/GED or Equivalent | 8(4–12) | 12(4–19) | 16(7–24) | 10(1–20) | 0.167 |
| | Some College or AA degree | 11(4–17) | 5(1–9) | 11(3–20) | 8(1–15) | 0.36 |
| | College Graduate or above | 5(2–8) | 5(1–11) | 4(1–9) | 7(1–12) | 0.059 |
| Marital Status | Married/Living with Partner | 9(3–14) | 10(1–18) | 9(1–18) | 9(2–15) | NA |
| | Never married | 27(10–45) | 10(2–22) | 15(2–32) | 13(1–27) | 0.218 |
| | Widowed/Divorced/Separated | 31(12–50) | 17(3–31) | 20(6–33) | 32(13–52) | 0.014 |
| Vigorous work activity | Yes | 10(2–21) | 8(1–15) | 7(1–15) | 3(1–7) | 0.276 |
| Moderate work activity | Yes | 12(3–22) | 6(1–11) | 9(2–15) | 9(3–15) | 0.42 |
| Hysterectomy | Yes | 27(17–36) | 15(2–27) | 19(8–30) | 25(8–42) | < .001 |
| Diabetes | Yes | 38(25–51) | 17(2–32) | 22(8–36) | 20(5–35) | < .001 |
| Asthma | Yes | 15(5–26) | 12(6–18) | 18(8–28) | 13(8–18) | 0.004 |
| Overweight | Yes | 13(7–19) | 6(2–11) | 14(7–21) | 12(5–20) | 0.044 |
| Arthritis | Yes | 23(15–31) | 19(12–25) | 24(11–37) | 21(10–33) | < .001 |
| Cancer | Yes | 25(12–37) | 42(12–72) | 7(8–21) | 33(13–52) | 0.054 |
| Depression | Yes | 12(8–16) | 11(6–15) | 15(7–23) | 17(6–29) | 0.0045 |

statistically significant difference between groups (P = 0.044). The prevalence of cancer increased non-significantly (P = 0.054) from 25% in 2011–2012 to 33% in 2017–2020 (Table 1).

Table 2 displays the relationships between key research variables and CVD in Black American women from 2011 to 2020. The following study factors were statistically significant: Being born in the US: Compared to women who were not born in the US, those who were born in the US had a 3.017 times higher risk of developing CVD. Education level: Compared to women with college degrees or above, women with less than a ninth-grade education had 0.213 times reduced odds of having CVD. Relationship status: Compared to women who were married or cohabiting, women who were widowed, divorced, or separated had 2.772 times higher odds of having CVD. Hysterectomy: Women who underwent a hysterectomy had a 2.879-times-higher risk of developing CVD than women who did not. Women who had diabetes, asthma, obesity, arthritis, and depression had increased probabilities of developing CVD than women who did not have these illnesses. The odds ratios were, correspondingly, 3.813, 1.918, 1.533, 5.575, and 1.718.

## Discussions

Using information from the NHANES database, this study analyzes the prevalence of five common cardiovascular disorders among adult Black American women, including congestive heart failure, coronary heart disease, heart attack, and stroke, as well as their related risk factors. Cardiovascular illnesses remain to be the leading cause of death and a major factor in

**Table 2. Association of CVD and study variables among Black American women from 2011–2020.**

| N = 3164029 (10.1%) | | OR | Lower Limit | Higher Limit |
|---|---|---|---|---|
| #Born in US | Yes | 3.017 | 0.959 | 9.493 |
| Education Level | Less Than 9th Grade | 0.213 * | 0.094 | 0.482 |
| | 9-11th Grade (Includes 12th grade with no diploma) | 0.694 | 0.409 | 1.177 |
| | High School Grad/GED or Equivalent | 0.526 | 0.125 | 2.219 |
| | Some College or AA degree | 0.493 | 0.243 | 1.001 |
| | College Graduate or above | Ref | Ref | Ref |
| Marital Status | Married/Living with Partner | Ref | Ref | Ref |
| | Never married | 1.776 | 0.700 | 4.506 |
| | Widowed/Divorced/Separated | 2.772 * | 1.243 | 6.184 |
| #Vigorous work activity | Yes | 0.700 | 0.364 | 1.347 |
| #Moderate work activity | Yes | 0.773 | 0.407 | 1.466 |
| #Hysterectomy | Yes | 2.879 * | 1.849 | 4.484 |
| #Diabetes | Yes | 3.813 * | 2.505 | 5.803 |
| #Asthma | Yes | 1.918 * | 1.252 | 2.939 |
| #Overweight | Yes | 1.533 * | 1.011 | |
| #Arthritis | Yes | 5.575 * | 2.776 | 11.198 |
| #Cancer | Yes | 2.351 | 0.981 | |
| #Depression | Yes | 1.718 * | 1.195 | 2.470 |

# "No" use as a reference

* statistically significant.

decreased life expectancy globally [29, 30]. 330 million years of life were lost as a result of cardiovascular illness in 2017, and an additional 35.6 million years were spent living with a handicap [29, 30]. More than 85.1% of all cardiovascular disease-related deaths in 2016 were attributable to ischemic heart disease and cerebrovascular disease (stroke) [31].

There was a substantial correlation between potential cardiovascular risk factors and the prevalence and variability of cardiovascular diseases among Black American women, according to a trend analysis carried out in each two-year survey cycle from 2011 to 2020. According to the data, Black American women with less than a ninth-grade education and those who had experienced divorce, widowhood, or separation were more likely to suffer from cardiovascular diseases. Additionally, it was discovered that among Black American women, hysterectomy, diabetes, asthma, depression, and cancer were all closely related to cardiovascular disease.

Women of Black heritage who were included in the trend analysis and had lower levels of education were more likely than other participants to have cardiovascular illnesses [32–37]. In line with findings from other research, the trend analysis also showed that divorced/widowed/separated women had a higher risk of CVD than married women [38–42]. In Black American women, hysterectomy was discovered to be a standalone risk factor for CVD, and CVD development was strongly correlated with diabetes. Although the link between hysterectomy and increased CAD risk is still unclear, some theories include low hormone status exacerbating atherosclerosis and reduced ovarian blood flow leading to early ovarian failure [43–50].

Despite the association between depression and cardiovascular diseases, few studies have looked at depression as a significant risk factor for the onset of CVD in gender-balanced sample groups [51]. However, during a 4.5-year follow-up period, Wassertheil-Smoller et al. (2004) found a significantly higher mortality risk for women with a 5-unit increase in depression score. Another study found a link between depressed symptoms and CVD mortality, but only in females [52]. In older women, but not in older males, depression was also discovered

to be an independent risk factor for heart failure [53]. According to [54], women experience depression at a rate that is double that of men, and being overweight or obese was linked to a significant risk of CVD events, particularly in women [55, 56]. According to several research [38, 57–59], a larger waist circumference increases the risk of CVD. In fact, Hadaegh et al. (2010) found waist circumference as one of the most important predictors of cardiovascular disease [59]. Nearly 50% of the increased risk for CVD has been shown to be mediated by obesity-related factors, including blood pressure categories, glucose intolerance status, and abnormal lipid profiles [38, 58].

Rheumatoid arthritis, for example, affects women disproportionately more than men and is linked to a higher risk of CVD [60, 61]. Additionally, according to our research, Black American women who have arthritis are also more likely to have cardiovascular diseases. Traditional risk assessments usually understate the likelihood of CVD in these populations; as a result, a number of professional associations have recently issued guidelines for risk assessment and primary CVD prevention in autoimmune illnesses such arthritis [60].

Additionally, earlier research [62, 63] has shown a link between asthma and a higher incidence of cardiovascular disease, particularly in women. These results are consistent with our hypothesis that Black American women with asthma had an increased risk of cardiovascular disease.

Since the study only covers the years 2011 through 2020, it may not be conceivable to identify long-term patterns and trends in CVD and medical conditions among Black American women. To better understand the changes in these health outcomes, future research might examine trends over a longer time span. Although, this study offers some insight into the relationship between CVD and different research variables, it is not evident whether all potential comorbidities and risk factors have been taken into account in the analysis. Additional risk factors and their influence on the incidence and prevalence of CVD and medical conditions among Black American women could be the subject of future research. Despite focusing on the prevalence of CVD and medical conditions among Black American women, the table does not discuss intervention strategies for these conditions. Further studies could look into potential obstacles to putting such interventions into practice as well as efficient interventions and management techniques for CVD and medical conditions in this population.

## Conclusions

There is a need for more comprehensive gender and ethnicity-based investigations into CVD, surgical procedures such as hysterectomy and NCDs including diabetes, asthma, depression, and arthritis, as well as the creation of gender-related biopsychosocial explanatory models. Gender bias is paramount to clinical and population health, necessitating future research. In conclusion, gender-related concerns must be considered not only in the detection of CVD but also in treatment and rehabilitation programs in order to fulfill the special needs of women better, hence enhancing the prevention of CVD.

## Acknowledgments

All credits go to Black American Health Disparity' Managing Director and Assistant Professor, George Mason University, Dr. Jennifer Warren.

## Author Contributions

**Conceptualization:** Rezaul Karim Ripon.

**Data curation:** Rezaul Karim Ripon, Umma Motahara.

**Formal analysis:** Rezaul Karim Ripon, Umma Motahara.

**Funding acquisition:** Umma Motahara.

**Investigation:** Rezaul Karim Ripon.

**Methodology:** Umma Motahara.

**Project administration:** Rezaul Karim Ripon, Md. Tajuddin Sikder.

**Software:** Rezaul Karim Ripon.

**Supervision:** Rezaul Karim Ripon, Shahriar Hossain.

**Validation:** Rezaul Karim Ripon.

**Visualization:** Rezaul Karim Ripon.

**Writing – original draft:** Rezaul Karim Ripon, Fahmida Hoque Rimti, Mickelder Kercy, Umma Motahara.

**Writing – review & editing:** Rezaul Karim Ripon, Shahriar Hossain, Umma Motahara, Md. Sharif Hossain, Md. Tajuddin Sikder.

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
