## [Decision Letter · Decision Letter 0]

3 Apr 2023

PGPH-D-22-01962

A Trend Analysis of African American Women with Cardiovascular Disease and Other Non-Communicable Chronic Medical Conditions from 2011 to 2020

Dear Dr. Ripon,

Thank you for submitting your manuscript to PLOS Global Public Health. After careful consideration, we feel that it has merit but does not fully meet PLOS Global Public Health’s publication criteria as it currently stands. Therefore, we invite you to submit a revised version of the manuscript that addresses the points raised during the review process.

We look forward to receiving your revised manuscript.

Kind regards,

Jianhong Zhou

Staff Editor

Journal Requirements:

1. Please send a completed 'Competing Interests' statement, including any COIs declared by your co-authors. If you have no competing interests to declare, please state "The authors have declared that no competing interests exist". Otherwise please declare all competing interests beginning with the statement "I have read the journal's policy and the authors of this manuscript have the following competing interests:"

3. Please provide separate figure files in .tif or .eps format only and remove any figures embedded in your manuscript file. Please also ensure that all files are under our size limit of 10MB.

4. We have noticed that you have uploaded Supporting Information files, but you have not included a list of legends. Please add a full list of legends for your Supporting Information files after the references list.

Additional Editor Comments (if provided):

Reviewers' comments:

Reviewer's Responses to Questions

**Comments to the Author**

1. Does this manuscript meet PLOS Global Public Health’s publication criteria? Is the manuscript technically sound, and do the data support the conclusions? The manuscript must describe methodologically and ethically rigorous research with conclusions that are appropriately drawn based on the data presented.

Reviewer #1: No

Reviewer #2: Partly

2. Has the statistical analysis been performed appropriately and rigorously?

Reviewer #1: No

Reviewer #2: No

3. Have the authors made all data underlying the findings in their manuscript fully available (please refer to the Data Availability Statement at the start of the manuscript PDF file)?

Reviewer #1: Yes

Reviewer #2: Yes

4. Is the manuscript presented in an intelligible fashion and written in standard English?

Reviewer #1: No

Reviewer #2: No

5. Review Comments to the Author

Reviewer #1: I do not think this paper has the required quality. The p value for trends are significant, but they are not well discussed, so their reasons are not known. Although the sample is called African American women, some are born outside the US, so this is probably Black not African American. Most national studies collect data on race (Black or African American) not ethnicity. So, I think this paper should be on Black women because it may have foreign born Black people too. Other than this, title is only on trends, but a major part of the paper is on correlates and risk factors as well. So, in fact, there are two main aims, not one aim.

Reviewer #2: Comment to Author

Abstract: Results in abstract are written very poor. for example; from 2nd line to 5th line''

what is the meaning of significant trend in CVD, author should have written some results, should have given some number like the prevalence of CVD has increased/decreased from XXX in 2009 to xxx in 2020.

from 3rd to 5th line: What does author wanted to say with this line?

"These women were more likely to report an education level less than 9th grade, their marital status as widowed/divorced/separated, and having a history of hysterectomy, diabetes, asthma, overweight, arthritis, and or depression."

Please rewrite a meaningful results.

Introduction: 1. There are very few arguments has been given in introduction section about the topic.

2. What is the rationale behind the study?

Methodology

1. Authors of this manuscript did not mentioned about data source. Methodology section directly started with study variables and directly jump to statistical analysis, which is not the proper method to write an methodology section of any paper.

Authors of this paper should have given a paragraph about th data source. Author need to write some information about data like, how data was collected and sampling design or where data is available.

2. what is meaning of this line in statistical analysis section: "We used weighted logistic regression to examine linear trends (crude) across survey cycles to determine P values for trends." I could not get it that logistic regression was used to determine p values for trends. what is the meaning of this line?

Results

line 133 to 135 "The prevalence of those with education level less than 9th grade". Does this line means that percentage of individuals with education level less than 9th grade....?

Author may want to write the prevalence of CVD among those with educational level 9th grade.....

Like this there are many results are not written very carefully whuile interpreting and creating confusion. All the results of this paper also written very poorly. I would suggest please rewrite again the result section.

Thank You

6. PLOS authors have the option to publish the peer review history of their article (what does this mean?). If published, this will include your full peer review and any attached files.

**Do you want your identity to be public for this peer review?** For information about this choice, including consent withdrawal, please see our Privacy Policy.

Reviewer #1: No

Reviewer #2: No

---

## [Decision Letter · Decision Letter 1]

28 Jul 2023

PGPH-D-22-01962R1

A Trend Analysis of Black American Women with Cardiovascular Disease and Chronic Medical Conditions, sociodemographic factors from 2011 to 2020

Dear Dr. Ripon,

Thank you for submitting your manuscript to PLOS Global Public Health. After careful consideration, we feel that it has merit but does not fully meet PLOS Global Public Health’s publication criteria as it currently stands. Therefore, we invite you to submit a revised version of the manuscript that addresses the points raised during the review process.

We look forward to receiving your revised manuscript.

Kind regards,

Jianhong Zhou

Staff Editor

Journal Requirements:

Additional Staff Editor Comments: the previous two reviewers are not available this time, therefore we invited a new reviewer (reviewer 3) for this revision.

Reviewers' comments:

Reviewer's Responses to Questions

**Comments to the Author**

1. If the authors have adequately addressed your comments raised in a previous round of review and you feel that this manuscript is now acceptable for publication, you may indicate that here to bypass the “Comments to the Author” section, enter your conflict of interest statement in the “Confidential to Editor” section, and submit your "Accept" recommendation.

Reviewer #3: All comments have been addressed

2. Does this manuscript meet PLOS Global Public Health’s publication criteria? Is the manuscript technically sound, and do the data support the conclusions? The manuscript must describe methodologically and ethically rigorous research with conclusions that are appropriately drawn based on the data presented.

Reviewer #3: Partly

3. Has the statistical analysis been performed appropriately and rigorously?

Reviewer #3: Yes

4. Have the authors made all data underlying the findings in their manuscript fully available (please refer to the Data Availability Statement at the start of the manuscript PDF file)?

Reviewer #3: Yes

5. Is the manuscript presented in an intelligible fashion and written in standard English?

Reviewer #3: Yes

6. Review Comments to the Author

Reviewer #3: This study uses NHANES data to look at how CVD burdens in Black women have changed between 2021 and 2020. The findings are meaningful and have potential clinical and policy implications.

Major issues:

1. Results: the authors started with Table 2. I cannot seem to find Table 1. Not sure if it was a typo or that they left out Table 1 by accident.

2. The authors should briefly mention by the other release cycles include data from two years, but the 2017-2020 included data for 4 years. Providing this kind of technical notes and details are important for contexts.

3. The missing data issues are complicated NHANES. Did the authors conduct any missing data analysis? Were the data missing at random? This is a major issue and needs to be addressed.

4. Line 162-163, the authors stated that they "conducted a 2-analysis using XXXX." Is this a typo? Not sure what is a "2-analysis." Also, to see there was a difference of what? Clarifications and more details are needed.

5. Figure 2: the title (both on top in red text and italicized texts on the bottom) is awkward phrased. Also, the y-axis in each plot did not start at 0, which skewed the actual magnitude of changes, i.e., making changes look bigger than they actually are. Also, 95% confidence intervals should be included to indicate whether the changes were significant.

6. Line 549: i assume the dot was a typo? Also, using on asterisk for statistical significance and two asterisks for reference group was confusing. I suggest they pick a different symbol for the reference group notation.

Minor issues:

1. I don't think it is necessary to include the actual variable names in the descriptions of the measurement. In fact i have never seen that done in studies using NHANES or any other secondary datasets. It is unnecessary, and might even be confusing and distracting to readers not familiar with NHANES datasets. I think they need to be removed.

2. Line 166, need to provide a citation for the R program. Line 160, need to include a citation for the NCHS policy and weighting instructions.

Overall, this article could benefit more careful proofreading and format editing.

Overall this revised manuscript is wanting a lot more careful proofreading and editing.

7. PLOS authors have the option to publish the peer review history of their article (what does this mean?). If published, this will include your full peer review and any attached files.

**Do you want your identity to be public for this peer review?** For information about this choice, including consent withdrawal, please see our Privacy Policy.

Reviewer #3: No

---

## [Decision Letter · Decision Letter 2]

4 Oct 2023

A Trend Analysis of Black American Women with Cardiovascular Disease and Chronic Medical Conditions, sociodemographic factors from 2011 to 2020

PGPH-D-22-01962R2

Dear Mr Ripon,

We are pleased to inform you that your manuscript 'A Trend Analysis of Black American Women with Cardiovascular Disease and Chronic Medical Conditions, sociodemographic factors from 2011 to 2020' has been provisionally accepted for publication in PLOS Global Public Health.

Best regards,

Priyanka Baloni

Academic Editor

Reviewer Comments (if any, and for reference):

Reviewer's Responses to Questions

**Comments to the Author**

1. If the authors have adequately addressed your comments raised in a previous round of review and you feel that this manuscript is now acceptable for publication, you may indicate that here to bypass the “Comments to the Author” section, enter your conflict of interest statement in the “Confidential to Editor” section, and submit your "Accept" recommendation.

Reviewer #3: All comments have been addressed

Reviewer #4: All comments have been addressed

2. Does this manuscript meet PLOS Global Public Health’s publication criteria? Is the manuscript technically sound, and do the data support the conclusions? The manuscript must describe methodologically and ethically rigorous research with conclusions that are appropriately drawn based on the data presented.

Reviewer #3: Yes

Reviewer #4: Yes

3. Has the statistical analysis been performed appropriately and rigorously?

Reviewer #3: Yes

Reviewer #4: Yes

4. Have the authors made all data underlying the findings in their manuscript fully available (please refer to the Data Availability Statement at the start of the manuscript PDF file)?

Reviewer #3: Yes

Reviewer #4: Yes

5. Is the manuscript presented in an intelligible fashion and written in standard English?

Reviewer #3: Yes

Reviewer #4: Yes

6. Review Comments to the Author

Reviewer #3: I believe the responses from the authors were adequate.

Reviewer #4: The authors have addressed the reviewer comments and have revised the manuscript significantly. The rationale and methods are now better written. While the authors touch on the Covid-19 pandemic and its impact on CVD in black American women, it would be helpful to add a few points in the discussion on the need to study Long Covid and association with CVD as well, or perhaps as a future goal.

7. PLOS authors have the option to publish the peer review history of their article (what does this mean?). If published, this will include your full peer review and any attached files.

**Do you want your identity to be public for this peer review?** For information about this choice, including consent withdrawal, please see our Privacy Policy.

Reviewer #3: **Yes: **Lin Zhu

Reviewer #4: **Yes: **Awanti Sambarey
